# Implicit Grasp Diffusion: Bridging the Gap between Dense Prediction and Sampling-based Grasping

**Pinhao Song[1], Pengteng Li[3], Renaud Detry[1,2]**
[1]KU Leuven, Dept. Mechanical Engineering, Research unit Robotics, Automation and Mechatronics
[2]KU Leuven, Dept. Electrical Engineering, Research unit Processing Speech and Images,
[3]HKUST (GZ), AI Thrust
{pinhao.song, renaud.detry}@kuleuven.be
pengteng.li@connect.hkust-gz.edu.cn

**Abstract:** There are two dominant approaches in modern robot grasp planning: dense prediction and sampling-based methods. Dense prediction calculates viable grasps across the robot's view but is limited to predicting a fixed number of grasps per voxel. Sampling-based methods, on the other hand, encode multi-modal grasp distributions, allowing for different grasp approaches at a point. However, these methods rely on a global latent representation, which struggles to represent the entire field of view, resulting in coarse grasps. To address this, we introduce *Implicit Grasp Diffusion* (IGD), which combines the strengths of both methods by using implicit neural representations to extract detailed local features and sampling grasps from diffusion models conditioned on these features. Evaluations on clutter removal tasks in both simulated and real-world environments show that IGD delivers high accuracy, noise resilience, and multi-modal grasp pose capabilities. Our code is freely available at https://gitlab.kuleuven.be/detry-lab/public/implicit-grasp-diffusion.git.

**Keywords:** Grasping, Implicit Neural Representations, Diffusion Models

## 1  Introduction

Grasping is the most basic activity that allows a robot to have an effect on its environment – by definition, a robot's primary purpose. Grasping is unfortunately an exceptionally challenging task. The robot must compute 6-DoF gripper poses that yield a stable bond with an object, based on incomplete visual information obtained from a single (color or depth) image of a cluttered scene, where at least half of object surfaces are self-occluded or occluded by neighboring objects. From this sparse view, the robot needs to understand the scene's geometry, object properties, and relationship to the shape of its gripper.

Today's solutions to this problem can be separated into two main categories: dense prediction methods [1, 2, 3] and sampling-based methods [4, 5]. For dense prediction methods, the model discretizes the scene in units (pixel [2], voxel [1], or point [3]), then generates a fixed number of grasps for each unit by regressing feasible gripper orientations at the unit's position, based on local geometry. By nature of this discretization, these methods can easily parse large scenes containing many objects. Unfortunately, a fixed number of grasp predictions per unit is a poor approximation of grasping: a point on an object can often be grasped via many gripper orientations. Capturing this multi-modality is essential to upstream processes that need to reconcile grasping with reachability or obstacle avoidance.

Sampling-based methods use heuristic rules [6, 7] or generative models [4, 5] to sample grasps. For the latter type, they encode a region of interest into a latent feature and use generative models such as VAEs or diffusion models to sample grasps within the region. In principle, this approach provides

8th Conference on Robot Learning (CoRL 2024), Munich, Germany.

strong multi-modality and a good approximation of the real world, since no discretization is applied. Unfortunately, they are expensive and difficult to train, and struggle with scenes that contain multiple objects: because the information encoded by the latent feature is global, when the region of interest contains multiple objects, the model misses local subtleties and generates inadequate grasp poses. Early works [4, 5] were limited to cases where a single object is shown in the scene. As shown in [8], the performance of SE(3)-DiffusionFields [5] achieves acceptable performance only after segmenting each object's point cloud and processing each object separately.

We propose to bridge the gap between the two approaches discussed above with a model named *Implicit Grasp Diffusion* (IGD), which captures strengths from both sides:

**(i) Local geometry:** Instead of encoding the entire scene into a global feature as sampling methods do, IGD leverages implicit neural representations [9] to query local features at certain positions. These local features prioritize local geometry information, which proves advantageous in generating valid grasps.

**(ii) Continuous sampling domain:** By contrast to the discretization operated by dense prediction methods, IGD samples grasps from a continuous domain, via implicit neural representations.

**(iii) Multi-modal representation:** We compute grasp orientations via diffusion at grasp locations, enabling the generation of multiple orientations at one location.

To realize these capabilities, we present two main technical contributions:

**1. Deformable attention in implicit space.** Feature relevance at a grasping point cannot be reduced to proximity alone. Instead, we utilize a Deformable Attention Module to dynamically sample the implicit feature space, determining relevance based on local geometry.

**2. Two-stage probabilistic grasp evaluator.** The large imbalance between valid and invalid grasps poses challenges for filtering the diffusion model's grasp suggestions. We propose a two-stage approach: an Affordance Evaluator followed by a Grasp Classifier. To capture grasp-relevant features, we introduce a grasp-conditioned Deformable Attention Module, enabling the model to maintain equivariance to 3D scene translations and rotations.

We conduct experiments on a clutter-removal task both in simulation and on physical hardware. IGD outperforms dense prediction methods (VGN [1], GIGA [10], GraspNet-1billion Baseline [11], and GSNet [12]) and sampling-based methods (GPD [6], 6DoF-GraspNet [4], and SE(3)-DiffusionField [5]).

## 2   Related Works

**6-Dof grasping methods.** Recent work on 6-Dof grasping has seen the emergence of two families of methods: sampling-based methods [4, 5, 6] and dense prediction methods [1, 3, 10]. In Section 1, we have already discussed the advantages and disadvantages of both frameworks. In summary, the locality and discretization of dense prediction methods reduce the training difficulty but sacrifice representational capability, while the globality and multi-modality of sampling-based methods lead to slow and poor convergence. In this work, the proposed IGD combines the complementary advantages of sampling-based and dense prediction methods. IGD focuses on local geometry and captures the multi-modal distribution of grasps, which makes it work well in cluttered scenes.

**Diffusion models.** Diffusion models [13] are trained to reverse the process of adding noise to data, effectively teaching the model to reconstruct the original data from progressively noisier versions through a series of denoising steps. Detailed explanations of diffusion models can be found in our supplementary material. Recent advances in computer vision [14, 15, 16] show diffusion models produce high-quality images without the drawbacks of mode collapse and unstable training. Due to its strong multi-modal expressiveness, this paradigm has also been applied to the robotics field to generate grasps [5], policies [17, 18], and trajectories [19, 20]. Our IGD model leverages diffusion

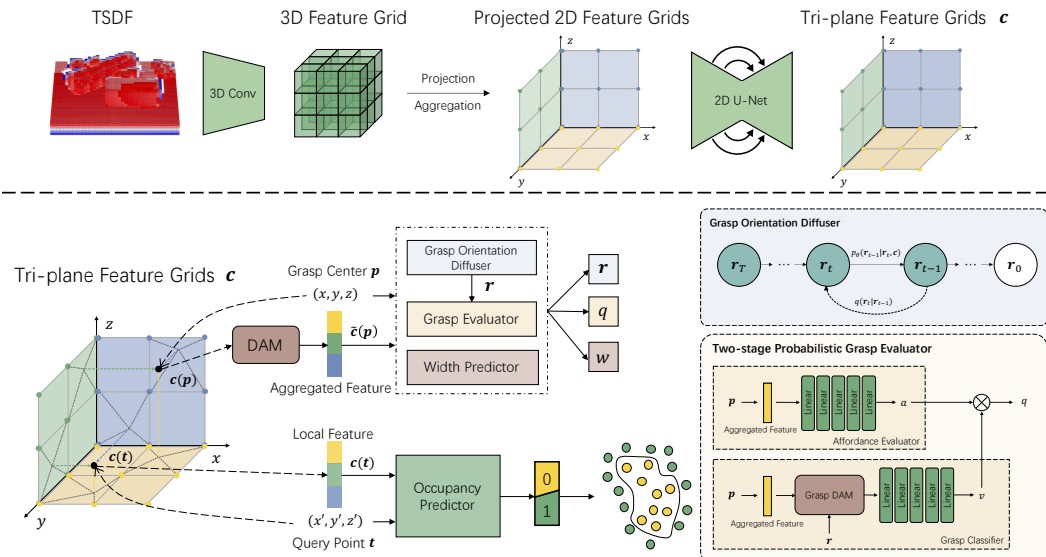

Figure 1: IGD workflow. The scene is captured by a TSDF obtained from a depth image. A 3D convolutional network extracts features from the TSDF, and the obtained features are projected onto three canonical planes, then aggregated into tri-plane feature grids. Then, a Grasp Orientation Diffuser samples grasp orientations conditioned on the aggregated feature queried at the grasp center. Finally, a Two-stage Probabilistic Grasp Evaluator estimates grasp quality.

to generate grasps, endowing it with an inherent capacity to probabilistically encode multiple hand-approach modes around one grasp location.

**Implicit Neural Representations.** Implicit neural representations (INRs) have shown remarkable capabilities in modeling 3D object shapes, synthesizing scene surfaces, and capturing complex structures [21, 22, 9]. INRs use MLPs to map spatial coordinates to scene attributes, enabling smooth and continuous representation of shapes in high resolution. INRs eliminate the need for discretization by effectively mapping continuous indices to corresponding data, such as magnifying images for super-resolution using a magnification scale as an index [23] or querying video frames in continuous time [24]. In robotics, INRs play a crucial role: [25, 26] use INRs to map gripper poses to grasp distances, guiding the gripper towards valid grasp poses, while GIGA [10] uses INRs to learn geometry and affordance jointly for acquiring grasp poses at query positions. In this work, the proposed IGD follows GIGA and uses INRs to eliminate spatial discretization. Combined with the multi-modal sampling of diffusion models, IGD can theoretically sample all the potential grasps in a workspace.

## 3 Implicit Grasp Diffusion

Implicit Grasp Diffusion (IGD) models the distribution $p(\boldsymbol{r}, w|\boldsymbol{p})$, capturing gripper orientations $\boldsymbol{r}$ and gripper widths $w$ conditioned on a grasp center point $\boldsymbol{p} \in \mathbb{R}^3$. As shown in Fig. 1, IGD comprises a Grasp Orientation Diffuser (GOD) that samples a grasp $\boldsymbol{g} = (\boldsymbol{p}, \boldsymbol{r})$ at $\boldsymbol{p}$, followed by a Grasp Evaluator that assesses the grasp quality $q \in [0, 1]$ of $\boldsymbol{g}$ and filters out low-quality grasps. Both models are based on a local feature representation of the object's geometry around $\boldsymbol{p}$, extracted by a Tri-plane Feature Encoder. A Width Predictor then estimates the finger width required for the grasp using the same aggregated feature.

### 3.1 Tri-plane Feature Encoder

Following GIGA [10], we adopt ConvONets [27] as the encoder architecture to extract a tri-plane feature space. The encoder processes a TSDF voxel field with a 3D CNN layer, generating a fea-

ture embedding for each voxel. These 3D features are then projected onto three canonical planes ($XY/XZ/YZ$). A 2D U-Net further refines these feature planes, producing the tri-plane features $c_{xy}, c_{xz}, c_{yz}$. Given a query position $p$, we project $p$ to each feature plane and query the local features at the projected locations $c(p)$, as:

$$c(p) = [\phi(c_{xy}, p_{xy}), \phi(c_{xz}, p_{xz}), \phi(c_{yz}, p_{yz})],  \tag{1}$$

where $c_{ij}, p_{ij}$ (i, j $\in$ x, y, z) are the plane features and points projected onto the corresponding plane, and $\phi$ represents a bilinear interpolation of the feature plane at the projected point.

To generate a grasp at position $p$, we gather features from $p$'s neighborhood by sampling nearby points and querying, then aggregating their features. Crucially, the sampling process is guided by local geometry to focus on areas relevant to the grasp. Inspired by [28], we propose a Deformable Attention Module (DAM, Fig. 2a) to dynamically adjust the receptive field and produce the aggregated feature $\tilde{c}(p)$, formulated as:

$$Q = W_Q c(p), \quad K = W_K \{c(p + \Delta p_k)\}_{k=1}^K, \quad V = W_V \{c(p + \Delta p_k)\}_{k=1}^K,  \tag{2}$$

$$\tilde{c}(p) = W_O V \, \text{Softmax}(\frac{K^T Q}{\sqrt{d_k}}),  \tag{3}$$

where $W_Q, W_K, W_V, W_O$ are learned parameters. $\Delta p_k$ is a learned offset obtained by linear projection from $c(p)$. In effect, $\tilde{c}(p)$ encodes a local shape descriptor tailored for grasp generation and grasp accuracy evaluation.

## 3.2 Grasp Orientation Diffuser

Given a position $p$, the Grasp Orientation Diffuser (GOD) models an orientation distribution $p_\theta(r|\tilde{c}(p))$ to represent feasible grasps based on the local geometry information encoded in $\tilde{c}(p)$. During the forward diffusion process, noise is injected into the orientation $r$ (denoted as $r_0$) to obtain $r_t$ as:

$$r_t = \sqrt{\bar{\alpha}_t} r_0 + (1 - \bar{\alpha}_t)\epsilon_t,  \tag{4}$$

where $\epsilon_t \in \mathcal{N}(0, I)$. In the reverse diffusion process, $\epsilon_t$ is approximated with $\epsilon_\theta(r, t)$ output by a noise predictor, which is realized through implicit neural representations, as:

$$f_\theta(g, \tilde{c}(p), t) \to \epsilon_\theta(r, t).  \tag{5}$$

where $f_\theta(\cdot)$ is MLPs with learned parameters $\theta$ and $g = (p, r)$. Thus, $r_0$ can be reconstructed from pure noise $r_T$ iteratively as follows:

$$\hat{r}_{t-1} = \frac{1}{\sqrt{\alpha_t}}(\hat{r}_t - \frac{1 - \alpha_t}{\sqrt{1 - \bar{\alpha}_t}}\epsilon_\theta(\hat{r}_t, t)) + \sigma_t z,  \tag{6}$$

where $z \in \mathcal{N}(0, I)$. Through the reverse diffusion process, the predicted grasp $\hat{g} = (p, \hat{r}_0) \in SE(3)$ can be obtained. To train the noise predictor, the KL divergence between the forward and backward diffusion processes is minimized, which is equivalent to:

$$\mathcal{L}_{ddpm} = \mathbb{E}_{t \sim [1,T], x_0 \sim q(r_0), \epsilon \sim \mathcal{N}(0,I)} \left[ ||\epsilon - \epsilon_\theta(\sqrt{\bar{\alpha}_t} r_0 + \sqrt{1 - \bar{\alpha}_t}\epsilon, t)||^2 \right].  \tag{7}$$

## 3.3 Two-stage Probabilistic Grasp Evaluator

Not all grasps generated by GOD are satisfactory: (i) Grasps generated far from any object, or inside one, should be discarded. (ii) Since grasps produced by GOD are samples, some have a low probability and are inadequate. Consequently, we employ a grasp evaluator to filter out unsatisfactory grasps. However, training a grasp evaluator directly is challenging due to the severe imbalance between graspable and ungraspable regions, a problem noted in similar works [4, 29, 30]. To address this, we propose a Two-stage Probabilistic Grasp Evaluator, designed to manage the imbalance between positive and negative grasps. We consider the conditional distribution $p(r|p)$ and rewrite it as:

$$p(r|p) = \sum_{a \in \{0,1\}} p(r|a, p)p(a|p),  \tag{8}$$

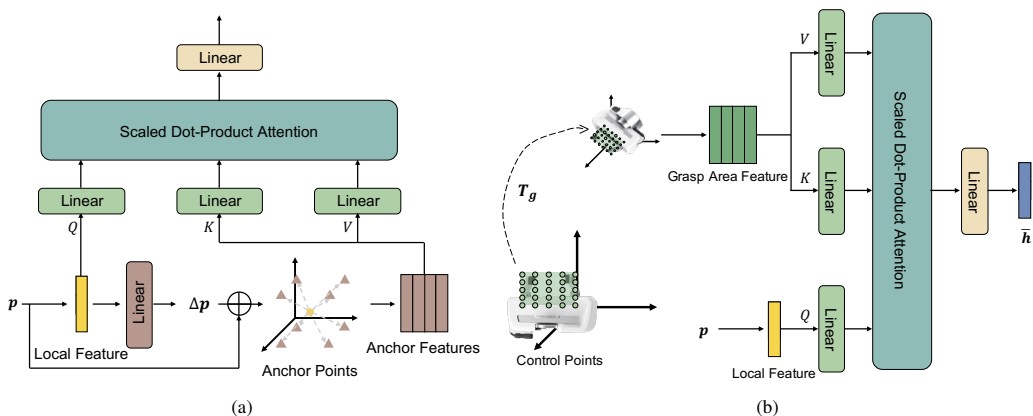

Figure 2: (a) The architecture of Deformable Attention Module. (b) The architecture of Grasp-conditioned Deformable Attention Module. See text for details.

where affordance $a$ represents the existence of feasible grasps at a certain point. $p(a|\boldsymbol{p})$ denotes the probability of the existence of feasible grasps at position $\boldsymbol{p}$, and $p(\boldsymbol{r}|a,\boldsymbol{p})$ denotes the feasibility probability of the grasp orientation $\boldsymbol{r}$ given the affordance $a$ at the position $\boldsymbol{p}$. Eq. 8 shows that the grasp evaluator can be separated into two factors: an Affordance Evaluator (AE) $p(a|\boldsymbol{p})$ and a Grasp Classifier (GC) $p(\boldsymbol{r}|a,\boldsymbol{p})$. To train the grasp evaluator, we maximize the lower bound of log-likelihood, which is an approximation of maximum likelihood estimation (MLE), as:

$$
\begin{aligned}
\log(p(\boldsymbol{r}|\boldsymbol{p})) &= \log(\sum_{a\in\{0,1\}} p(\boldsymbol{r}|a,\boldsymbol{p})p(a|\boldsymbol{p})), \\
&\geq \sum_{a\in\{0,1\}} \log(p(\boldsymbol{r}|a,\boldsymbol{p})) + \log(p(a|\boldsymbol{p})) \text{ (Jensen's inequality)}
\end{aligned}
\tag{9}
$$

The lower bound of the log-likelihood can be regarded as a summation of the log-likelihood of AE and GC. Maximizing the lower bound reduces the original goal to independent maximum-likelihood objectives for the first and second stages respectively. In our implementation, implicit neural representations are used to realize AE and GC.

**Affordance Evaluator (AE).** We query the grasp center $\boldsymbol{p}$ and the aggregated feature $\tilde{\boldsymbol{c}}(\boldsymbol{p})$ to obtain the affordance, as:

$$
f_\psi(\boldsymbol{p}, \tilde{\boldsymbol{c}}(\boldsymbol{p})) \to a \in [0,1]\,,
\tag{10}
$$

where $f_\psi$ is implemented by several residual fully-connected blocks. AE is trained with the following cross-entropy loss:

$$
\mathcal{L}_a = -(a_{\boldsymbol{p}} \log(\hat{a}_{\boldsymbol{p}}) + (1-a_{\boldsymbol{p}}) \log(1-\hat{a}_{\boldsymbol{p}})),
\tag{11}
$$

where $\hat{a}_{\boldsymbol{p}}$ is the predicted affordance, while $a_{\boldsymbol{p}}$ is the ground-truth affordance.

**Grasp Classifier (GC).** We query the grasp $\boldsymbol{g}$ and the aggregated feature $\tilde{\boldsymbol{c}}(\boldsymbol{p})$ to obtain the grasp score $v$, as:

$$
f_\varphi(\boldsymbol{g}, \tilde{\boldsymbol{c}}(\boldsymbol{p})) \to v \in [0,1]\,.
\tag{12}
$$

To implement Eq. 12, we propose a Grasp-conditioned Deformable Attention Module (Grasp DAM, illustrated in Fig. 2b) to extract a feature for grasp classification. We first define a set of gripper-relative learnable control points $\boldsymbol{u}_1, ..., \boldsymbol{u}_L$. In order to express these points in the scene's base frame, we transform them with a transformation $T_{\boldsymbol{g}}$ defined by $\boldsymbol{g}$. The feature for grasp classification can be obtained by applying deformable attention to the features of control points, as:

$$
\boldsymbol{Q}' = \boldsymbol{W}_Q' \boldsymbol{c}(\boldsymbol{p}), \quad \boldsymbol{K}' = \boldsymbol{W}_K' \{\boldsymbol{c}(T_{\boldsymbol{g}}\boldsymbol{u}_l)\}_{l=1}^L, \quad \boldsymbol{V}' = \boldsymbol{W}_V' \{\boldsymbol{c}(T_{\boldsymbol{g}}\boldsymbol{u}_l)\}_{l=1}^L,
\tag{13}
$$

$$
\bar{\boldsymbol{h}} = \boldsymbol{W}_O' \boldsymbol{V}' \text{ Softmax}(\frac{\boldsymbol{K}'^T \boldsymbol{Q}'}{\sqrt{d_k}}),
\tag{14}
$$

where $\boldsymbol{W}'_Q, \boldsymbol{W}'_K, \boldsymbol{W}'_V, \boldsymbol{W}'_O$ are learned parameters. Several residual fully connected blocks are applied to $\bar{\boldsymbol{h}}$ to compute a grasp score $v$. However, due to the sparsity of feasible grasps, GC trained with cross-entropy on a balanced positive-negative dataset often overestimates the quality of infeasible grasps. To alleviate this problem, a large set of negative grasps are randomly sampled from the scene and a focal loss is used to extract valuable information from these negative examples, as:

$$\mathcal{L}_g(\hat{v}_{\boldsymbol{g}}) = \begin{cases} -\alpha(1 - \hat{v}_{\boldsymbol{g}})^\gamma \log(\hat{v}_{\boldsymbol{g}}), & v_{\boldsymbol{g}} = 1, \\ -(1 - \alpha)\hat{v}_{\boldsymbol{g}} \log(1 - \hat{v}_{\boldsymbol{g}}), & v_{\boldsymbol{g}} = 0, \end{cases} \tag{15}$$

where $\hat{v}_{\boldsymbol{g}}$ is the predicted grasp score, while $v_{\boldsymbol{g}}$ is the ground-truth grasp label. $\alpha$ is the balance parameter, and $\gamma$ is the focus parameter. The sampling of control points in GraspDAM is $SE(3)$-equivariant because if the geometric relation between the scene and grasp remains fixed, the relation between transformed control points and the scene will be consistent. This contrasts with a straight-forward alternative implementation, where sampling points are generated by neural networks with the input of the grasp and aggregated feature. The sampling equivariant limits the search space and improves the grasp feature extraction consistency, which improves the training stability. Based on Eq. 8, $p(\boldsymbol{r}|\boldsymbol{p}) = p(\boldsymbol{r}|a = 1, \boldsymbol{p})p(a = 1|\boldsymbol{p})$ as $p(\boldsymbol{r}|a = 0, \boldsymbol{p}) = 0$. Thus, we can obtain the final grasp quality as $q = a \cdot v$. In the paper's supplementary material, ablation studies demonstrate that the Two-stage Probabilistic Grasp Evaluator has benefits in both training and inference.

### 3.4 Network Training

The training loss of the proposed IGD consists of two parts: the grasp loss and the geometry loss, as $\mathcal{L} = \mathcal{L}_{\text{grasp}} + \mathcal{L}_{\text{occ}}$. The total grasp loss can be formulated as $\mathcal{L}_{\text{grasp}} = \mathcal{L}_{\text{ddpm}} + \mathcal{L}_a + \mathcal{L}_g + \mathcal{L}_w$, where $\mathcal{L}_{\text{ddpm}}$, $\mathcal{L}_a$, and $\mathcal{L}_g$ are the loss for training the GOD, AE, and GC, respectively. The width loss $\mathcal{L}_w$ and the geometry loss $\mathcal{L}_{\text{occ}}$ are obtained by calculating the L2 distance and the cross-entropy loss between predictions and ground truths, respectively, following the paradigm of GIGA [10].

## 4 Experiments

### 4.1 Experimental Setup

We follow the same experimental setup as GIGA [10]. The model is trained with GIGA's open-source data [10]. We implement our method on a Franka Research 3 robot arm in both simulation and real-world environments.

**Simulation Environment:** Our simulated environment is built using PyBullet, featuring a free-floating gripper that samples grasps within a tabletop workspace measuring $30 \times 30 \times 30\,\text{cm}^3$. For a fair comparison, we employ the same object assets as VGN [1] and GIGA [10], including 303 training and 40 test objects from various datasets [31, 32, 33, 34]. The model is validated in two types of simulated scenes: *pile* and *packed*. In the pile scene, objects are randomly dropped into a box of the same dimensions as the workspace, resulting in a cluttered pile once the box is removed. The packed scene features a subset of taller objects placed at random locations on the table in their canonical pose. A single-view depth map serves as the model's input for grasp generation. The process involves repeatedly predicting and executing a grasp, followed by removing the grasped object from the workspace until one of three conditions is met: all objects are cleared, two consecutive failures occur, or no grasp is detected. Performance metrics are averaged over 100 simulation rounds using 5 different random seeds.

**Real-world Environment:** In real-world experiments, 15 rounds of experiments are performed for both the packed and pile scenes, respectively. Everyday objects are used to conduct the experiments (see supplementary material). In each round, 5 objects are randomly selected and placed on the table. In each grasp trial, we pass the TSDF or point cloud from a side-view depth camera to the model and execute the physically feasible grasp with the highest score.

Table 1: Quantitative results of clutter removal. We report the mean and standard deviation of GSR and DR. $N$ denotes sampling rounds in IGD. The best performances are highlighted in bold. Our method shows equal or better performance than all other methods, with competitive latency.

| Method | Packed | | Pile | | Latency (ms) |
|---|---|---|---|---|---|
| | GSR (%) | DR (%) | GSR (%) | DR (%) | |
| VGN [1] | 72.5±2.6 | 76.7±1.7 | 59.3±2.9 | 43.5±2.9 | **5** |
| GIGA [10] | 84.8±2.2 | 85.1±2.5 | 69.5±1.3 | 49.0±3.4 | 17 |
| GraspNet-1B Baseline [11] | 49.9±2.3 | 40.1±2.2 | 50.2±4.2 | 30.0±2.3 | 73 |
| GSNet [12] | 67.8±2.5 | 60.1±3.2 | 58.3±3.8 | 51.3±4.6 | 149 |
| GPD [6] | 41.8 ± 2.9 | 34.1±3.4 | 22.7±1.1 | 9.0±0.7 | 2138 |
| 6DoF-GraspNet [4] | 17.9±0.8 | 11.9±0.9 | 15.5±2.9 | 6.9±1.1 | 2220 |
| SE(3)-Dif [5] | 7.2±1.5 | 4.3±1.0 | 7.6±1.8 | 3.0±0.8 | 5643 |
| IGD (Ours, $N$=1) | **92.9±1.8** | 86.7±1.8 | 68.2±1.9 | 50.6±1.5 | 217 |
| IGD (Ours, $N$=11) | 91.2±0.9 | **88.8±1.5** | **71.8±2.2** | **55.7±2.6** | 1823 |

Table 2: Quantitative results of clutter removal in the real-world experiment. We report GSR, DR, successful grasp numbers, and total grasp trial numbers (in brackets). The best performances are highlighted in bold. Our method outperforms VGN and GIGA on both scenes and both metrics.

| Method | Packed | | Pile | |
|---|---|---|---|---|
| | GSR (%) | DR (%) | GSR (%) | DR (%) |
| VGN [1] | 77.3 (58/71) | 81.7 | 65.3 (47/72) | 62.7 |
| GIGA [10] | 81.3 (65/80) | 86.7 | 77.4 (65/84) | 86.7 |
| IGD (Ours) | **88.3 (68/77)** | **90.7** | **82.7 (67/81)** | **89.3** |

**Metric:** (i) Grasp Success Rate (GSR = $\frac{\#\text{successful grasps}}{\#\text{total grasps}}$) that measures the ratio of successful grasps to total grasps; (ii) Declutter Rate (DR = $\frac{\#\text{grasped objects}}{\#\text{total objects}}$) that measures the ratio of objects removed successfully to the number of total objects presented.

## 4.2 Training and Inference Details

We implement the proposed IGD with *PyTorch* and train the models with the *AdamW* optimizer for 12 epochs. An initial learning rate of $2 \times 10^{-4}$ is set. The step learning scheduler is leveraged with a decay factor set to $0.1$, and the scheduler works at the $9^{\text{th}}$ and $11^{\text{th}}$ epochs.

The final grasp pose is obtained by sampling from the trained IGD. We discretize the volume of the workspace into $40 \times 40 \times 40$ voxel grids and use the centers of all voxel cells as grasp centers. We then evaluate affordances at all grasp centers, filtering out those with low affordances. Then, for each remaining grasp center, we conduct multi-round grasp sampling and retain the grasp with the highest grasp score. Next, we mask out impractical grasps. Finally, grasp qualities are the product of affordances and grasp scores, and the grasp with the highest quality is selected if the quality is beyond the threshold. If no grasp has a quality above the threshold, we declare that there is no feasible grasp in the scene.

## 4.3 Baselines

We compare our method against seven strong baselines:

**Dense Prediction Methods**: **(i) VGN** [1]: Volumetric Grasping Network, which generates a large number of grasps in parallel given input TSDF volume. **(ii) GIGA** [10]: Grasp detection via Implicit Geometry and Affordance, which leverages implicit neural representations and geometrical supervision. **(iii) GraspNet-1billion Baseline** [11]: A baseline method training on the GraspNet-1billion dataset. **(iv) GSNet** [12]: A model based on GraspNet-1billion Baseline with the proposed graspness as a refined grasp quality label.

**Sampling-based Methods**: **(v) GPD** [6]: Grasp Pose Detection, which generates a large set of grasp candidates and classifies each of them. **(vi) 6DoF-GraspNet** [4]: A grasping model based on VAE. **(vii) SE(3)-Dif** [5]: A grasping model based on SE(3) score-based diffusion models.

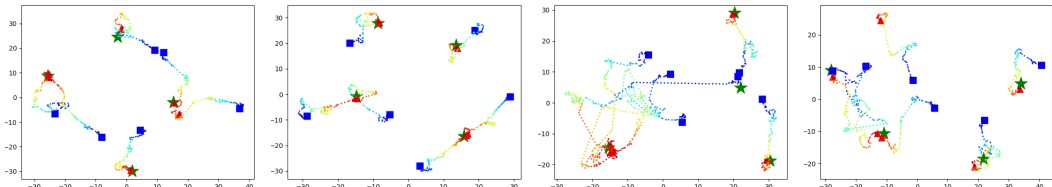

Figure 3: The visualization of the denoising trajectory. Blue squares are the starting points of the denoising process, while red triangles are the endpoints. Green stars are ground-truth grasps. Starting from pure noise, the data can converge to different ground truths during the denoising process, demonstrating IGD's multi-modal expressiveness.

For all the methods mentioned above, we use their pre-trained models for comparison.

### 4.4 Grasp Detection Results

We report GSR and DR for different scenes in Table 1. Sampling-based methods struggle to generate effective grasps in cluttered environments, performing worse than dense prediction methods. IGD outperforms both dense prediction and sampling-based methods with just one sampling round in both packed and pile scenes. In packed scenes, IGD shows an 8.1% improvement in GSR over GIGA while maintaining the same DR level. Increasing the sampling rounds to 11 boosts IGD's DR in packed scenes to 88.8%, exceeding GIGA by 3.7%. In pile scenes, IGD's performance matches that of GIGA, and with 11 sampling rounds, it achieves 71.8% GSR and 55.7% DR. The superior performance of IGD results from its integration of the strengths of both sampling-based and dense prediction methods: (i) local information enables effective handling of cluttered scenes, and (ii) the strong multi-modal expressiveness of the diffusion model captures grasp distributions. Table 2 presents grasp detection results from real-world experiments, aligning with simulation findings where IGD outperforms GIGA in both packed and pile scenes. This enhanced performance is partly attributed to IGD's greater robustness in noisy real-world environments, which will be analyzed further in the supplementary material.

### 4.5 Multi-modal Grasp Modeling

The introduction of diffusion models aims to capture the multi-modal distribution of feasible grasps. To validate this, we conduct six rounds of diffusion processes at various valid points containing feasible grasps and visualize the denoising trajectories using t-SNE, as shown in Fig. 3. Each figure displays four ground-truth grasps. The results indicate that starting from pure noise, the data converges to different ground truths throughout the denoising process, demonstrating that IGD effectively captures the multi-modal nature of viable grasps.

## 5 Conclusion

In this work, we introduce Implicit Grasp Diffusion (IGD), a novel grasping framework that leverages the strengths of both dense prediction and sampling-based methods, exhibiting robust locality and multi-modality. Our framework features a Grasp Orientation Diffuser and a Two-stage Probabilistic Grasp Evaluator for regressing and evaluating grasps. We assess IGD in both simulated and real-world environments, comparing it against state-of-the-art approaches. Experimental results highlight IGD's high grasp accuracy, strong noise robustness, and expressive multi-modal capabilities.

**Acknowledgments**

This work is supported by Interne Fondsen KU Leuven/Internal Funds KU Leuven. We would like to thank Yang Chen at Shenzhen BIT-MSU University for providing computational resources and a working environment.

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

# Supplementary Material of Implicit Grasp Diffusion

**Pinhao Song[1], Pengteng Li[3], Renaud Detry[1,2]**

[1]KU Leuven, Dept. Mechanical Engineering, Research unit Robotics, Automation and Mechatronics
[2]KU Leuven, Dept. Electrical Engineering, Research unit Processing Speech and Images,
[3]HKUST (GZ), AI Thrust
{pinhao.song, renaud.detry}@kuleuven.be
pengteng.li@connect.hkust-gz.edu.cn

## 1  Preliminary: Diffusion Models

Diffusion models aim to model an unknown data distribution $q(\boldsymbol{x}_0)$ with a parameterized model $p_\theta(\boldsymbol{x}_0)$. The procedure consists of two steps: the forward and the reverse diffusion processes. The forward process iteratively injects small Gaussian noise in $\boldsymbol{x}_0$ to obtain $\boldsymbol{x}_{1:T}$:

$$q(\boldsymbol{x}_{1:T}|\boldsymbol{x}_0) = \prod_{t=1}^{T} q(\boldsymbol{x}_t|\boldsymbol{x}_{t-1}), \tag{1}$$

where $q(\boldsymbol{x}_t|\boldsymbol{x}_{t-1}) = \mathcal{N}(\boldsymbol{x}_t; \sqrt{1-\beta_t}\boldsymbol{x}_{t-1}, \beta_t\boldsymbol{I})$ is the per-step noise injection following variance schedule $\beta_1, ..., \beta_T$. This leads to the distribution $q(\boldsymbol{x}_t|\boldsymbol{x}_0) = \mathcal{N}(\boldsymbol{x}_t; \sqrt{\bar{\alpha}_t}\boldsymbol{x}_0, (1-\bar{\alpha}_t)\boldsymbol{I})$, where $\alpha_t = 1 - \beta_t$ and $\bar{\alpha}_t = \prod_{i=1}^{t} \alpha_i$. Since $\bar{\alpha}_t \approx 0$, $\boldsymbol{x}_T \sim \mathcal{N}(0, \boldsymbol{I})$. The reverse diffusion learns to denoise the data starting from $\boldsymbol{x}_T$ following $p_\theta(\boldsymbol{x}_{t-1}|\boldsymbol{x}_t) = \mathcal{N}(\boldsymbol{x}_{t-1}; \mu_\theta(\boldsymbol{x}_t, t), \beta_t\boldsymbol{I})$ where:

$$\mu_\theta(\boldsymbol{x}_t, t) = \frac{1}{\sqrt{\alpha_t}}(\boldsymbol{x}_t - \frac{\beta_t}{\sqrt{1-\bar{\alpha}_t}}\boldsymbol{\epsilon}_\theta(\boldsymbol{x}_t, t)). \tag{2}$$

The parameterized model $\boldsymbol{\epsilon}_\theta(\boldsymbol{x}_t, t)$ is called the score function, and it is trained to predict the perturbations and the noising schedule by the score-matching objective:

$$\arg\min_\theta \mathbb{E}_{t\sim[1,T],\boldsymbol{x}_0\sim q,\boldsymbol{\epsilon}\sim\mathcal{N}(0,\boldsymbol{I})} \left[||\boldsymbol{\epsilon} - \boldsymbol{\epsilon}_\theta(\sqrt{\bar{\alpha}_t}\boldsymbol{x}_0 + \sqrt{1-\bar{\alpha}_t}\boldsymbol{\epsilon}, t)||^2\right]. \tag{3}$$

In particular, such a score function represents the gradient of the learned probability distribution as:

$$\nabla_{\boldsymbol{x}_t} \log p_\theta(\boldsymbol{x}_t) = -\frac{1}{\sqrt{1-\bar{\alpha}_t}}\boldsymbol{\epsilon}_\theta(\boldsymbol{x}_t, t). \tag{4}$$

## 2  Experiment Scenes

The objects used in the real-world experiment are shown in Fig. 1a. Fig. 1b and Fig. 1c illustrate examples of packed and pile scenes.

## 3  Visualization of Grasp Detection

We visualize the top-10-score grasps with a threshold of 0.5 in some challenging cases in Fig. 2. Compared to GIGA, IGD generates more collision-free good-quality grasps and gives lower scores to bad-quality grasps.

We also visualize some failure cases of IGD in Fig. 3. There are three main reasons for these failures. First, using diffusion models allows for sampling a large range of orientations, which increases the burden on the grasp evaluator. When the range of feasible orientations is limited, such as with large objects (Fig. 3 (a-b)), smooth curved surfaces (Fig. 3 (c-d)), and lying boxes (Fig. 3 (d)), the proposed method is prone to failure. Second, as a single-view method based on TSDF generated

8th Conference on Robot Learning (CoRL 2024), Munich, Germany.

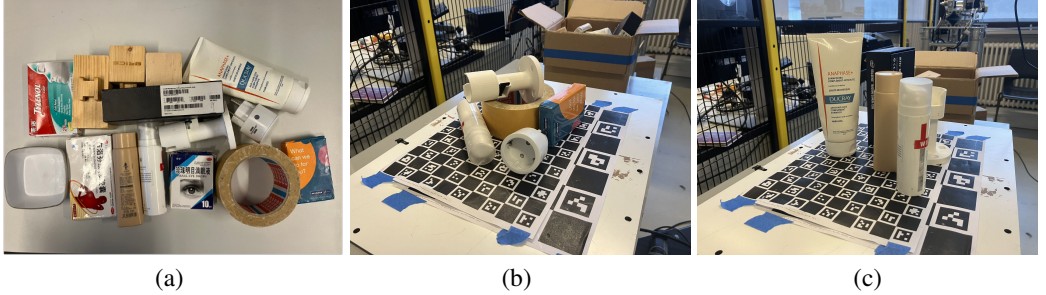

(a)                    (b)                    (c)

Figure 1: (a) Objects for the real-world declutter experiment. (b) An illustration of the pile scene. (c) An illustration of the packed scene.

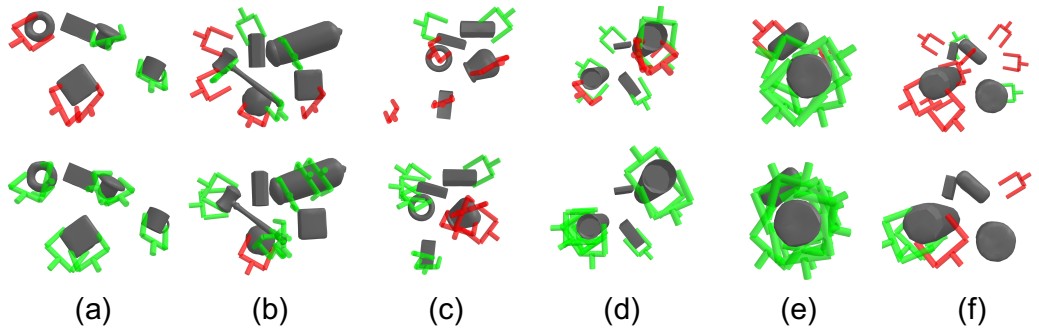

(a)       (b)       (c)       (d)       (e)       (f)

Figure 2: Grasp visualization in some challenging cases. The first row denotes GIGA, and the second row denotes IGD. (a-c) are in pile scenes, while (d-f) illustrate packed scenes. Green grasps denote successful grasps, while red grasps denote failed grasps.

from a depth map, it struggles to analyze the geometry and borders of occluded objects (as shown in Fig. 3 (e-f)), leading to further failures. Third, IGD uses a low-resolution TSDF without the guidance of RGB information, resulting in a loss of subtle geometry details. For example, in Fig. 3 (g), IGD generates grasps between two objects, incorrectly assuming they are connected. In Fig. 3 (h), IGD fails to distinguish between thin-lying objects and the table, resulting in an unsuccessful grasp attempt.

## 4 Robustness to Different Noises

Table 1 shows the performance of GIGA and IGD in different kinds of noises. According to [1], noise in a depth camera includes the following noises: noise from stereo matching, lateral noise, blur, and noise in depth estimation. We simulate those noises and add them in the depth image to evaluate the performance of models. Our baseline condition is the dex noise environment, where models are trained. According to Table 1, both IGD($N$=1) and IGD($N$=9) outperform GIGA in different noise conditions. Besides, the performance decrease of IGD is also lower than GIGA. The results demonstrate the strong robustness of IGD. The robustness of IGD comes from two factors: (i) Probabilistic two-stage grasp evaluator can precisely estimate the true quality of grasps; (ii) Multi-round sampling increases the chance to sample good grasps. From Table 1, increasing sampling rounds decreases performance drop. Due to its strong robustness, IGD performs well in real-world experiments.

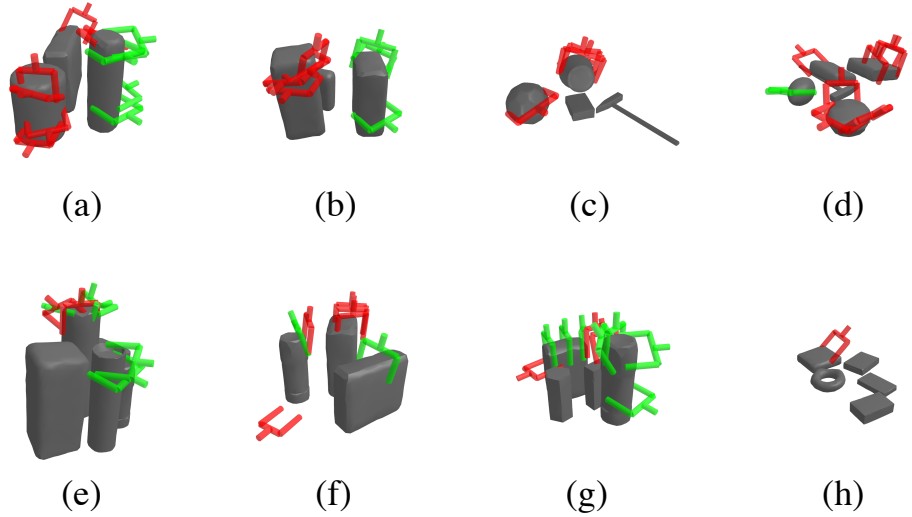

Figure 3: Different failure cases of IGD.

Table 1: Robustness experiments. We evaluate the performance under four types of noises: dex, stereo, lateral, blur, and depth. $N$ denotes sampling rounds in IGD. We set dex noise as the baseline to calculate the performance decrease (in bracket). The best performances are highlighted in red.

| Noise Type | Method | Packed | | Pile | |
|---|---|---|---|---|---|
| | | GSR (%) | DR (%) | GSR (%) | DR (%) |
| Dex (baseline) | GIGA [2] | 84.8±2.2 | 85.1±2.5 | 69.5±1.3 | 49.0±3.4 |
| | IGD ($N$=1) | 92.9±1.8 | 86.7±1.8 | 68.2±1.9 | 50.6±1.5 |
| | IGD ($N$=9) | 91.2±0.9 | 88.8±1.5 | 71.0±0.7 | 55.0±1.6 |
| Stereo | GIGA [2] | 72.0±1.6 (↓ 12.8) | 78.4±2.6 (↓ 6.7) | 51.8±1.2 (↓ 17.7) | 47.0±2.3 (↓ 2.0) |
| | IGD ($N$=1) | 83.6±1.8 (↓ 9.3) | 86.4±2.1 (↓ 0.3) | 60.4±0.9 (↓ 7.8) | 48.6±1.6 (↓ 2.0) |
| | IGD ($N$=9) | 85.6±1.3 (↓ 5.6) | 88.3±1.2 (↓ 0.5) | 61.9±1.5 (↓ 9.1) | 54.8±2.5 (↓ 0.2) |
| Lateral | GIGA [2] | 75.1±1.4 (↓ 9.7) | 81.5±1.0 (↓ 3.6) | 56.3±1.5 (↓ 13.2) | 53.8±2.8 (↑ 4.8) |
| | IGD ($N$=1) | 83.7±1.2 (↓ 9.2) | 86.4±1.8 (↓ 0.3) | 63.6±1.5 (↓ 4.6) | 53.2±2.4 (↑ 2.6) |
| | IGD ($N$=9) | 85.2±1.6 (↓ 6.0) | 87.2±1.4 (↓ 1.6) | 68.6±2.9 (↓ 2.4) | 61.2±2.2 (↑ 6.2) |
| Blur | GIGA [2] | 69.6±2.4 (↓ 15.2) | 72.4±1.2 (↓ 12.7) | 53.8±1.9 (↓ 15.7) | 48.2±2.7 (↓ 0.8) |
| | IGD ($N$=1) | 81.9±1.7 (↓ 11.0) | 79.8±2.2 (↓ 12.7) | 60.9±2.9 (↓ 7.3) | 43.5±3.1 (↓ 7.1) |
| | IGD ($N$=9) | 84.2±1.5 (↓ 7.0) | 84.5±1.7 (↓ 4.3) | 63.4±2.4 (↓ 7.6) | 48.4±3.0 (↓ 6.6) |
| Depth | GIGA [2] | 75.3±1.7 (↓ 9.5) | 83.6±1.2 (↓ 1.5) | 51.8±1.8 (↓ 17.7) | 47.6±2.1 (↓ 1.4) |
| | IGD ($N$=1) | 89.9±0.6 (↓ 3.0) | 88.6±1.4 (↑ 1.9) | 58.5±0.8 (↓ 9.7) | 49.2±1.4 (↓ 1.4) |
| | IGD ($N$=9) | 90.0±1.1 (↓ 1.2) | 90.1±0.5 (↑ 1.3) | 62.1±1.3 (↓ 8.9) | 56.5±3.1 (↑ 1.5) |

## 5 Ablation Studies

We also conducted extensive ablation studies to validate each module in our proposed IGD. Ablation studies were mainly performed in pile scenes because of data abundance and higher task difficulty compared to packed scenes. Table 2 shows the ablation studies of IGD. The proposed DAM effectively improves the performance in both GSR and DR. Besides, the performance deteriorates with GC alone compared to the AE. Combining the AE and GC achieves the best performance, which demonstrates the effectiveness of the proposed probabilistic two-stage grasp evaluator. To further investigate what brings the performance improvement, we train AE and GC together and deactivate one of them in the inference (shown as "GC*" and "AE*" in Table 2). If we deactivate the GC, the performance decreases drastically, while the performance decline from deactivating the AE is limited. This result is contrary to solely training the AE and GC. We can conclude that the performance improvement is mainly from the loss supervision instead of a simple ensemble of two modules. Negative grasp sampling is also important to train the grasp evaluator. Introducing neg-

Table 2: Ablation Studies of the proposed IGD. "DAM" denotes the deformable attention module. "AE" denotes the affordance evaluator. "GC" denotes the grasp classifier. "Neg." denotes the negative grasp sampling for the grasp classifier. "AE*" and "GC*" denote that AE and GC are trained together, but they are used solely in the inference. The best performances are highlighted in red.

| DAM | AE | GC | Neg. | AE* | GC* | GSR (%) | DR (%) |
|---|---|---|---|---|---|---|---|
|  | ✓ |  |  |  |  | 59.1±2.8 | 42.8±3.1 |
| ✓ | ✓ |  |  |  |  | 62.2±3.0 | 45.6±2.5 |
| ✓ |  | ✓ |  |  |  | 53.5±2.6 | 42.5±4.0 |
| ✓ |  | ✓ | ✓ |  |  | 72.2±1.8 | 33.5±2.0 |
| ✓ | ✓ | ✓ |  |  |  | 62.8±3.1 | 47.9±4.3 |
| ✓ | ✓ | ✓ | ✓ |  |  | 68.2±1.9 | 50.6±1.5 |
| ✓ |  |  | ✓ | ✓ |  | 59.5±0.9 | 47.5±0.9 |
| ✓ |  |  | ✓ |  | ✓ | 64.3±2.3 | 48.0±3.1 |

Table 3: Ablation Studies of anchor points in DAM. The best performances are highlighted in red.

| Anchor points | GSR (%) | DR (%) |
|---|---|---|
| $2^3$ | 68.2±1.9 | 50.2±1.5 |
| $3^3$ | 65.7±2.3 | 49.0±2.7 |
| $4^4$ | 62.3±1.9 | 46.2±2.5 |

Table 4: Ablation Studies of focal loss. The best performances are highlighted in red.

| $\gamma$ | $\alpha$ | GSR (%) | DR (%) |
|---|---|---|---|
| 0 | 0.5 | 68.5±2.9 | 46.1±3.9 |
| 1 | 0.25 | 65.7±1.9 | 43.6±1.4 |
| 1 | 0.5 | 66.5±1.8 | 49.7±1.4 |
| 1 | 0.75 | 63.2±2.6 | 47.5±2.6 |
| 2 | 0.25 | 66.3±2.1 | 46.9±3.1 |
| 2 | 0.5 | 68.2±1.9 | 50.6±1.5 |
| 2 | 0.75 | 65.2±3.5 | 50.4±2.9 |
| 3 | 0.5 | 62.7±3.5 | 45.8±3.3 |

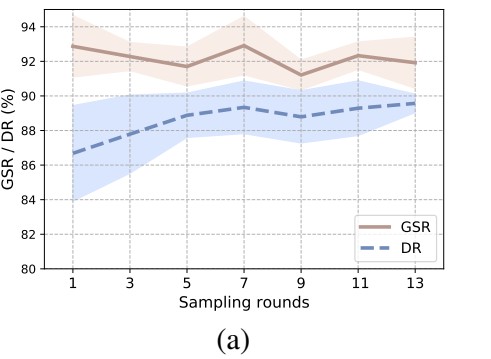

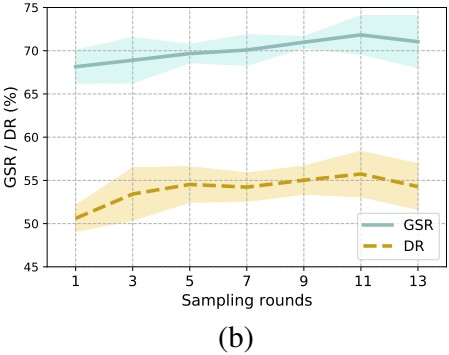

Figure 4: The ablation study of sampling rounds. (a) Packed scene. (b) Pile scene.

ative grasp sampling with GC alone largely increases GSR (53.5% to 72.2%) but decreases DR (42.5% to 33.5%). When it comes to probabilistic two-stage structure, negative sampling improves both GSR and DR. The results mean that the grasp evaluator needs to excavate information from negative samples to learn to distinguish feasible grasps in the large grasp space.

**Number of sampling rounds.** Since we can sample multiple rounds to obtain different grasps at the same grasp position, an ablation study about sampling rounds is conducted to analyze the effect of this hyper-parameter, which is shown in Fig. 4. In packed scenes, there is no improvement in GSR

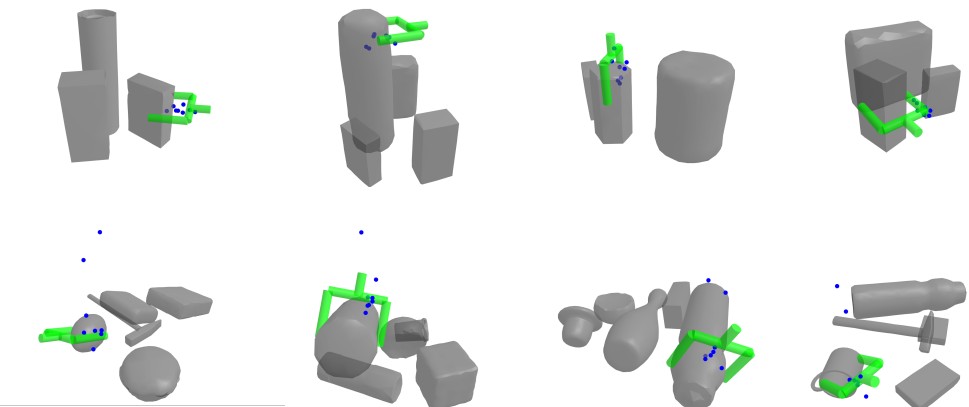

Figure 5: The visualization of the sampled points $\boldsymbol{p} + \Delta\boldsymbol{p}_k$ (shown in blue) for the grasp (shown in green) at position $\boldsymbol{p}$. The first column denotes the packed scene, and the second column denotes the pile scene.

as sampling rounds increase, while DR increases in general. In contrast, in harder pile scenes, both GSR and DR increase as sampling rounds increase. The results show the ability of IGD to obtain good grasps by increasing grasp samples, which is inherited from sampling-based methods.

**Sampled points in DAM.** Table 3 shows an ablation study on sampled points in DAM. Because the anchor points are initialized as the points spatial-uniformly sampled in the cube around the grasp center, we select $2^3$, $3^3$, and $4^3$ sampled point numbers to evaluate their performance. From Table 3, the best performance appears in the $2^3$ setting. We also visualize the sampled points in DAM in Fig. 5. In packed scenes, the sampled points are primarily distributed in the region between the grasping pose and the object's surface. Since grasps are generated by the features aggregated by DAM, we believe these sampled features contribute to more accurate grasp generation. In pile scenes, in addition to points localized around the grasp area, some points are distributed over a farther range. This visualization demonstrates that DAM can dynamically sample features based on the local geometry, aiding in accurate grasp generation.

**Focal loss hyper-parameters.** Table 3 shows an ablation study of focal loss hyper-parameters in GC. $\gamma = 0$ reduces the focal loss to a normal cross-entropy loss and shows inferior performance. $\gamma$ is to tune the level of hard example mining, and $\alpha$ is to balance the weight of positive and negative samples. According to the results, we select the best hyper-parameter setting as $\alpha = 0.5$ and $\gamma = 2$.

## 6 Limitations and Future Work

Although we have demonstrated the effectiveness of IGD in both simulation and real-world systems, there are limitations that future work can improve. First, we can't directly obtain the point with the highest affordance value. In GIGA, $40 \times 40 \times 40$ points are sampled to query the grasp to obtain the best grasp. IGD inherits this limitation from GIGA. Second, diffusion models have higher computational costs and inference latency compared to dense prediction methods, especially in IGD where grasp sampling is separated into two stages: position sampling and orientation sampling. Future work can exploit the latest advancements in diffusion model acceleration methods to reduce the number of inference steps required, such as new noisy schedules [3], inference solvers [4], and consistency models [5]. Third, in our GOD, we directly apply diffusion models to the generation of quaternions. However, quaternion in the diffusion process is not in close form. Current works have proposed a lot of methods to achieve rotation diffusion [6, 7, 8]. Although our effort to apply these works to IGD fails, it is still worth exploring.

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
