# OpenReview forum: "Implicit Grasp Diffusion: Bridging the Gap between Dense Prediction and Sampling-based Grasping"
_robot-learning.org/CoRL/2024/Conference — CoRL 2024_

### Official Review · Reviewer_TcPf · 2024-07-15

**Originality:** 2
**Technical Quality:** 2
**Clarity Of Presentation:** 3
**Potential Impact:** 2
**Recommendation:** 3
**Confidence:** 3

**Review:**

**Strengths**

1. The method demonstrates a performance improvement over GIGA.
2. The deformable-based local feature aggregation proves effective in predicting grasps, as validated through experiments.

**Weaknesses**

1. **Argumentation Issues:**
    - The categorization of grasping methods into dense and sampling-based is unconvincing.
    - The designation of Contact-GraspNet as a sampling-based method is questionable, as it not only does not utilize generative models, but also use descretized unit as points to predict a single grasp per point.
    - Previous works addressing the uni-modality in approach direction are not mentioned or compared [1-3]. These studies have effectively managed uni-modality while preserving local geometry, so the authors should have claims against them.
2. **Lack of Original Contributions:**
    - The approach heavily relies on previous work, particularly GIGA’s architecture and methods.
    - Claims of novelty in two-stage evaluation are undermined by prior implementations of similar methods, which are not cited [1-4].

References:

[1] GraspNet-1Billion: A Large-Scale Benchmark for General Object Grasping

[2] Graspness Discovery in Clutters for Fast and Accurate Grasp Detection

[3] Grasping as Inference: Reactive Grasping in Heavily Cluttered Environment

[4] Deep Learning Approaches to Grasp Synthesis: A Review

**Quality Of The Limitations Section:**

2

**Questions For Rebuttal:**

**Additional Concerns**

1. **Technical and Expository Clarifications Needed:**
    - Lack of thorough investigation of related works, particularly missing citations for ConvONet in 3.1.
    - Several terms and symbols (e.g., \phi_p) are undefined.
    - Is your f actually mapping? isn’t this a probabilistic model?
    - In Affordance Evaluator box of Figure 1, is usage of local feature correct? Isn’t it aggregated local feature?
    - Term Grasp diffuser is confusing. It only generates orientations.
    - Methods for generating negative grasps and determining ground truth for affordance are not adequately explained.
    - The voxel resolution for position sampling needs specification.
2. **Efficiency Concerns:**
    - Increased latency in grasp generation should be analyzed, identifying the main bottlenecks, possibly related to the grasp diffuser.

**Robotics Focus:**

4

**Summary Of Paper:**

The paper presents an extension of the GIGA algorithm, incorporating deformable feature aggregation, a two-stage probabilistic grasp evaluator, and diffusion-based orientation proposals. The enhanced method generates a tri-plane feature grid by processing TSDF with ConvONet, applies deformable heads for predicting grasp affordance and width, and employs a diffusion model for orientation generation. The authors demonstrate improvements over GIGA in both simulated and real-world grasping experiments.

**Summary Of Recommendation:**

I recommand weak accept.

---

### Official Review · Reviewer_bjQA · 2024-07-20
**The paper proposes a novel method and ideas that would be useful to the community. A few additional technical details and improvements for clarity are required.**

**Originality:** 3
**Technical Quality:** 4
**Clarity Of Presentation:** 4
**Potential Impact:** 3
**Recommendation:** 3
**Confidence:** 4

**Review:**

**Strengths**:
* The paper is well written and very well motivated from the start, making it easier to read.

* Combining strengths from dense prediction and sampling-based methods using a two-stage grasp evaluator and a diffusion process is a nice idea.

* The motivation behind using learnt offsets and control points to aggregate local information using the proposed Deformable Attention Module and not rely on a single set of features is strong.

* Experiments are well conducted and look convincing.

**Weaknesses**:

* The method still relies on the discretisation of the workspace to sample the grasp points. It should be addressed early in the paper, otherwise, the reader is led to believe that the proposed approach gets rid of the discretisation.
* It is not clear why the proposed Deformable Attention Module is not used in the diffusion process to generate orientation samples. Does the feature of the grasp point hold enough information to accurately predict the rotation?
* It is not clear how the learned offset \delta p_k is generated. How do you ensure that generated offsets are different from each other? Why not use the same control points as for the grasp-conditioned Deformable Attention Module with a fixed orientation?
* The claim that the Grasp Classifier is SE(3)-invariant is not explained properly. Implicit Neural Representation used has no such property and the features of the control points will change if the observation is rotated. It seems that a set of features of the control points is just an over-parametrization of SO(3).
* Figure 1 doesn’t clearly convey the procedure at inference time. I would suggest labelling different parts of Figure 1 (bottom) to clearly show the control flow at inference.
* It is not clear what is shown in Figure 3. Diffusion is used for sampling the rotation only but visualisation shows a diffusion process in a 2D plane.
* More information about the representation of the rotation, normalisation and noise scheduling strategies are needed.
* It would be good if limitations were moved from the appendix to the main paper.

**Quality Of The Limitations Section:**

2

**Questions For Rebuttal:**

* Why Deformable Attention Module is not used in the diffusion process to generate orientation samples?
* How the learned offset \delta p_k is generated?  How do you ensure that generated offsets are different from each other? Why not use the same control points as for the grasp-conditioned Deformable Attention Module with a fixed orientation?
* What makes the Grasp Classifier SE(3)-invariant?
* What rotation representation are you using and how do you normalise it during the diffusion process?
* What is actually shown in Figure 3?

If my concerns expressed in the limitations are addressed and the questions above answered, I’m willing to improve my score.

**Robotics Focus:**

4

**Summary Of Paper:**

This paper proposes a new method for grasp generation using Implicit Neural Representations, a two-stage grasp evaluator and a diffusion process for calculating the orientation of the grasp. The proposed approach uses features from the Implicit Neural Representation together with the proposed Deformable Attention Module to estimate the affordance of sampled points and the grasp score when rotation is computed using a diffusion process. In such a way, the method combines the advantages of using dense prediction and sampling-based methods. In the experiments, the proposed approach is compared against several baselines showing promising performance.

**Summary Of Recommendation:**

I would recommend accepting this paper for publication given that the authors add a few additional technical details and improvements for clarity.

---

### Official Review · Reviewer_cuMq · 2024-07-24

**Originality:** 3
**Technical Quality:** 4
**Clarity Of Presentation:** 3
**Potential Impact:** 3
**Recommendation:** 3
**Confidence:** 5

**Review:**

# Quality

This paper is of medium-high quality. The key intuitions behind the approach are sound - performing a global search that must yield a locally precise distribution of results (especially when the search space is over SE(3)) is a challenging task, and the design decisions chosen to address this are logical and well-reasoned.

The design of the Deformable Attention Module makes sense, and plausibly captures a dynamic context depending on query point and local geometry. Since this is a novel module, I would have liked to see some analysis of what geometric features are actually sampled, qualitatively. I don’t expect an interprable set, but in practice what regions of the space are sampled? Is attention hyperlocal or long-ranging? I’m somewhat skeptical that the formulation for predicting the delta offsets is sufficiently flexible for their desired effect, but am amenable to it.

In particular, I appreciate the decomposition of grasp scoring into an affordance term and a quality term, as it is a natural graphical model for the space. This allows them to address the class imbalance problem rather elegantly.

The diffusion process for rotations seems suspect, because rotations are not additive and therefore the diffusion trajectory may contain invalid rotations (depending on representation). The authors should take a look at the diffusion process for SE(3) in the “RPDiff” paper (https://anthonysimeonov.github.io/rpdiff-multi-modal/), in which they propose a modified compositional diffusion of elements of so3.

The “invariance” argument for the grasp region attention is tenuous - the features need not be invariant; it is simply the distribution over sampled points on the gripper which are invariant (or really, equivariant). Perhaps I misunderstood something.


# Clarity

The paper is mostly clear - however there are two major issues i had while reading:

1. In the method introduction, Equation 1 implies a regression rather than a generative approach, which confused me until I read further. Probably should be made clear that f maps from a point to a distribution around that point, from which you can sample and then score.
2. What was the significance of the occupancy grid? It’s not mentioned how it’s trained or why it’s used.

# Originality

The paper is of medium originality. Architecturally parts are quite similar to prior work and work in other domains, as is the decomposition into affordance+quality scoring (although the design, including focal loss, is principled). Additionally, while the attention module is novel, the design is not fully explored.

# Significance

The paper has some significance - I think that the design decisions are reasonable and lead to good performance, and while the final performance is incrementally better than prior work I expect that there is a lot of room for architectural improvements  in the design of the attention module + network which could be explored by the community / in future work.

# Strengths

* The design of the network and inference algorithm is principled and intuitive
* The attention module is novel and clever, if a bit underexplored
* The phrasing of the problem as an implicit scoring problem w/ a diffusion-based generator makes sense
* Method outperforms baselines, on identical tasks

# Weaknesses
* Attention module is underexplored
* Improvements are modest
* Failure modes not adequately charactreized

**Quality Of The Limitations Section:**

2

**Questions For Rebuttal:**

One area that is not well-motivated is the Tri-plane feature grid. Why was this chosen, and are there benefits to simply using a 3D voxel grid?

Can you characterize behavior of the attention modules a bit more?

What are the failure modes of the grasping? What leads to poor predictions in some cases?

**Robotics Focus:**

4

**Summary Of Paper:**

In this work, the authors consider the task of SE(3) grasp prediction. Noting the difficulty of capturing the multimodaltiy, precision, and sparsity of grasps in a real scene, the authors seek to combine both dense prediction and local sampling approaches in order to fit the underlying distribution. The authors propose a novel deformable attention module to effectively capture local features near a query point, and incorporate this attention module into various implicit functions which are used to sample and score grasps. The authors show that their method outperforms the next closest method, GIGA, as well as various other recent methods for grasp generation.

**Summary Of Recommendation:**

Overall there is enough novelty in the method - and downstream improvement over baselines - to warrant acceptance. If more details were explored, and failure modes characterized, I might be willing to increase my rating.

---

### Author Rebuttal · Authors · 2024-08-09

We thank the reviewers for their insightful comments and valuable feedback. First of all, we are encouraged by the reviewers' appreciation of the design of Two-stage probabilistic Grasp Evaluator and DAM. The reviewers point out the unclarity of rotation diffusion, SE(3)-invariance, and categorization, which helps improve the clarity of this paper. The advice on visualizing sampled points helps demonstrate the mechanism of DAM.

We uploaded the response to all the reviewers in the attachment. A revised paper and a supplementary material are also in the attachment with the revised part highlighted in blue.

---

### Decision · Program_Chairs · 2024-09-04

**Decision:**

Accept

**Comment:**

The reviewers generally agree that this is a valuable contribution with some novel ideas and well-demonstrated performance improvements. On the other hand, all three point out weaknesses in terminology and formal description (regarding Eqn. 1, SE(3) invariance, generation/sampling/discretization). There are also questions around the attention module and about limitations/failures.

The authors carefully addressed most of the reviewer's questions to their satisfaction.